# Weak Neutrino (Antineutrino) Charged-Current Responses and Scaling for Nuclear Matter in the Relativistic Mean Field

Sara Cruz-Barrios [1,*] , Guillermo D. Megias [2,*] and Juan A. Caballero [2]

[1] Departamento de Física Aplicada I, Universidad de Sevilla, 41080 Sevilla, Spain

[2] Departamento de Física Atómica, Molecular y Nuclear, Universidad de Sevilla, 41080 Sevilla, Spain; jac@us.es

[*] Correspondence: sara@us.es (S.C.-B.); megias@us.es (G.D.M.)

**Abstract:** A systematic analysis of the weak responses for charged-current quasielastic neutrino-nucleus reactions is presented within the scheme of a fully relativistic microscopic model considering momentum-dependent scalar and vector mean field potentials in both the initial and final nucleon states. The responses obtained are compared with the ones corresponding to simpler approaches: energy-independent potentials and the relativistic plane wave limit in the final state, i.e., no potentials applied to the outgoing particle. The analysis is also extended to the scaling phenomenon, which provides additional information regarding nuclear dynamics. Results for the scaling function are shown for various nuclei and different values of the transferred momentum in order to analyze the behavior of the relativistic scalar and vector mean field potentials.

**Keywords:** neutrino interactions; electroweak interactions; nuclear matter; scaling; relativistic mean field





## 1. Introduction

The assessment of neutrino oscillation experiments [1–6], which are of great relevance to measure the leptonic charge-parity (CP) violation phase, determine the neutrino mass hierarchy, and strengthen the current picture of the oscillation mixing angles, requires an accurate description of how neutrinos interact with complex nuclear systems, as they constitute the main ingredients in the detectors. In recent years, different groups have devoted great effort to this problem using alternative descriptions of the reaction mechanism and nuclear modeling, also including different kinematical regions where reaction channels from low to very high momentum transfer can play a significant role [7–20]. This makes a crucial difference from electron scattering experiments, where the electron beam energy is known with high accuracy. On the contrary, in the case of neutrino beams, their energy can range from a region extended from tens of MeV to several GeV [1,2,21].

From the analysis of quasielastic (QE) electron scattering data, emerges the phenomenon of scaling and superscaling, that is, at high enough values of the transfer momentum, $q$, the differential cross section divided by an appropriate single-nucleon response shows a tiny dependence with $q$, and is the same for all nuclear systems. This has been explored at depth in several works [8,22–28], where not only scaling/superscaling is clearly fulfilled, but also the scaling function extracted from the data shows a clear asymmetric shape with a tail extended to high values of the transfer energy, $\omega$ [23,29]. Our group has been involved in this problem for the last few years and has developed various theoretical descriptions based on the behavior of data [8,26,27] and incorporating relativistic mean field (RMF) effects [30–32]. The RMF is one of the few microscopic approaches able to reproduce the asymmetric shape of the phenomenological scaling functions, also producing an enhancement in the transverse components, an effect due to the relativistic dynamics incorporated in the lower components of the nucleon wave functions. Despite

its undoubted success, not only for electrons but also when extended to neutrino processes [26,30,31], the RMF model clearly fails at very high values of $q$ due to the very strong energy-independent scalar and vector potentials involved in the final states. This shortfall was remedied with the SuSAv2 (SuperScaling Approach version 2) model [8,26], which incorporates not only the RMF responses, but also results based on the relativistic plane wave impulse approximation (RPWIA). The SuSAv2, originally restricted to the QE region, has been extended by incorporating 2p-2h meson exchange currents (MEC) [27,33] and effects related to the region of deep inelastic scattering (DIS) [34]. SuSAv2 is presently used by most experimentalists in the analysis of neutrino scattering data [35–38] corresponding to very different collaborations: MiniBooNE, MINERvA, MicroBooNE, T2K, DUNE, etc.

In this work, we do not intend to provide a precise description of the data. Our interest is focused on the limits of the RMF and, in particular, its failure to behave properly at high values of the transfer momentum, $q$. We then follow the same strategy as used in our previous work devoted to electron scattering [39]. Here, we extend the study to neutrino processes providing general expressions for the weak nuclear responses in several approaches. Our analysis is entirely based on the use of the RMF applied to nuclear matter. Clearly, this is an oversimplified description of the process with respect to more sophisticated RMF approaches [30,31], but it allows us to test the specific role played by the different ingredients of the model, with special emphasis on the scalar (S) and vector (V) potentials used in the initial and final nucleon states. We show results for the five weak responses corresponding to different options for the potentials; from the most general case with $S$ and $V$ dependent on energy and different for initial and final states (EDSV model), to some simplified cases where only the potentials enter in the initial state (RPWIA approach), and $S$ and $V$ are constant and equal in the initial and final states (CtSV model). Finally, we also investigate scaling/superscaling properties by evaluating the scaling function at different kinematics ($q$-values) and for various nuclear systems (carbon, oxygen, and calcium). This provides an answer on how scaling of the first and second kinds works for each particular description of the relativistic potentials. It is worth remarking that the nuclear targets employed in this analysis are of relevance for current and forthcoming neutrino experiments, where carbon and oxygen are part of the T2K, MINERvA, SuperK, and HyperK detectors and calcium is similar to argon, which is employed in MicroBooNE or DUNE.

This manuscript is organized as follows: The theoretical scheme of the work is given in Section 2, where the basic formalism and the general expressions related to the weak charged current neutrino response functions within the context of the relativistic mean field model applied to nuclear matterit is described. The scaling and superscaling properties of these nuclear responses are also carefully analyzed. In Section 3, the outcomes for the weak longitudinal and transverse response functions corresponding to very different kinematic situations and nuclear targets are shown. A thouroug discussion of the scaling functions obtained within the previous schems is also carried out. Finally, in Section 4 we outline the main conclusions of this research.

## 2. General Formalism

In this work we restrict ourselves to inclusive charged-current quasielastic (CCQE) (anti)neutrino-nucleus scattering processes. In the laboratory frame and assuming the Born approximation, i.e., one virtual charged boson exchanged and leptons described as free particles, the differential cross section in terms of the transfer energy ($\omega$) and the lepton solid angle ($\Omega_\ell$) can be written in the general form:

$$\frac{d\sigma}{d\Omega_\ell d\omega} = \sigma_0 \mathcal{F}_\chi^2 \,, \tag{1}$$

where the index $\chi = +(-)$ refers to neutrino (antineutrino) processes and the $\sigma_0$ term depends on the leptonic variables and the weak couplings. Its general expression is given as:

$$\sigma_0 = \frac{G_F^2 \cos \theta_C^2}{2\pi^2} k_\ell E_\ell \cos^2 \frac{\widetilde{\theta}}{2} f_{rec}^{-1} \tag{2}$$

with $\theta_C$ the Cabibbo angle, $G_F$ the Fermi weak coupling constant, $k_\ell$ ($E_\ell$) the momentum (energy) of the final lepton, $f_{rec}$ the recoil factor, and $\widetilde{\theta}$ a generalized scattering angle defined as $\tan^2(\widetilde{\theta}/2) \equiv |Q^2|/v_0$ with $v_0 = 4E_\nu E_\ell - |Q^2|$. The terms $E_\nu$ and $Q^2$ represent the incident (anti)neutrino energy and the four-momentum transferred in the proces.

The whole information on the nuclear structure is contained in $\mathcal{F}_\chi^2$, given in terms of the weak response functions, $R_K$:

$$\mathcal{F}_\chi^2 = V_{LL}R_{LL} + V_{CC}R_{CC} + 2V_{CL}R_{CL} + V_T R_T + \chi V_{T'}R_{T'} , \tag{3}$$

where the kinematical factors, $V_K$, depend only on the lepton variables, and their explicit expressions can be found in [21]. The notation $C$, $L$, $T$, and $T'$ refer to charged (0), longitudinal (3), and transverse components (1, 2) with respect to the direction of the momentum transfer **q** (3rd component). This decomposition can be clearly observed in the nuclear response functions.

These weak nuclear response functions, $R_K$, are given from the corresponding components of the polarization propagator $\Pi^{\mu\nu}$ (also named as the current-current correlation function). Contrary to the case of electron scattering, where only the time component enters in the longitudinal response due to current conservation, here one needs to evaluate separately the time and longitudinal components:

$$R_{CC} = -\frac{2\mathcal{N}}{\pi\rho}\text{Im}\left\{\Pi^{00}\right\} \tag{4}$$

$$R_{LL} = -\frac{2\mathcal{N}}{\pi\rho}\text{Im}\left\{\Pi^{33}\right\} \tag{5}$$

$$R_{CL} = \frac{\mathcal{N}}{\pi\rho}\text{Im}\left\{\Pi^{03} + \Pi^{30}\right\} \tag{6}$$

$$R_T = -\frac{2\mathcal{N}}{\pi\rho}\text{Im}\left\{\Pi^{11} + \Pi^{22}\right\} \tag{7}$$

$$R_{T'} = \frac{i\mathcal{N}}{\pi\rho}\text{Im}\left\{\Pi^{12} - \Pi^{21}\right\} , \tag{8}$$

where $\mathcal{N}$ represents the number of protons (neutrons) in the nuclear target contributing to antineutrino (neutrino) scattering processes. We use a coordinate system with the $z$(3)-axis in the direction of **q**. The $T'$ term corresponds to the vector-axial interference in the polarization propagator, which is only present in weak interactions.

The weak nuclear responses (4)–(8) are calculated in a local density approximation from nuclear matter, where the density is defined as $\rho = 2k_F^3/(3\pi^2)$ and $k_F$ is the Fermi momentum. Because closure can be applied to perform the sum over all final states, the polarization propagator can be described in terms of the full propagator of the nuclear many-body system:

$$\Pi^{\mu\nu}(q,\omega) = -i \int \frac{d^4P}{(4\pi)^4} Tr[\hat{G}(P+Q)\widetilde{\Gamma}^\mu \hat{G}(P)\widetilde{\Gamma}^\nu] , \tag{9}$$

where $\hat{G}(P)$ represents the Green function for a nucleon propagator, as defined in [39], and $\widetilde{\Gamma}^\mu$ is the weak vertex given by:

$$\widetilde{\Gamma}^\mu = F_1^V \gamma^\mu + F_2^V \frac{i\sigma^{\mu\nu}Q_\nu}{2M} + G_A \gamma^\mu \gamma^5 + F_P Q^\mu \gamma^5 . \tag{10}$$

While $F_1^V$ and $F_2^V$ are related to the vector components of the hadronic current, $G_A$ and $F_P$ correspond to the axial terms. The axial form factor, $G_A$, describes the axial-pseudovector structure of the nucleon, while $F_P$ englobes its pseudoscalar contribution. Moreover, due to isospin symmetry, the isovector form factors, $F_1^V$ and $F_2^V$, can be simply related to the electromagnetic Dirac, $F_1$, and Pauli, $F_2$, form factors for protons and/or neutrons. In this paper, the electromagnetic proton and neutron form factors used correspond to the well-known Galster parametrization [40]. The term $M$ represents the mass of the nucleon. The axial form factor, $G_A$, is parametrized using a dipole form [41,42], with $g_A = -1.267$ as the axial-vector coupling constant and $M_A = 1.032$ GeV as the nucleon axial mass. Finally, the pseudoscalar form factor, $G_P$, is connected with the axial one, making use of the PCAC (partially conserved axial current) hypothesis [43].

The single-nucleon wave functions are described as solutions of the Dirac equation in the presence of relativistic scalar, $S$, and vector, $V$, potentials that may include dependence on the energy-momentum:

$$[\boldsymbol{\alpha} \cdot \mathbf{p} + \beta(M + S(\mathbf{p})) + V(\mathbf{p})]\psi(\mathbf{p}) = E_{\mathbf{p}}\psi(\mathbf{p}),\tag{11}$$

where $\boldsymbol{\alpha}$ and $\beta$ are the Dirac matrices and $E_{\mathbf{p}} = \sqrt{\mathbf{p}^2 + M^2}$. As shown, while the vector term is directly linked to the energy of the particle, the scalar one modifies its mass. In what follows, we introduce the effective nucleon four-momentum. Notice that this applies to both the initial and final nucleon states determined by their momenta, $\mathbf{p}_i$ and $\mathbf{p}_f$. For clarity, we simply use a generic notation $\mathbf{p}$:

$$P^{*\mu} = (p^0 - V(\mathbf{p}), \mathbf{p}) = (E_{\mathbf{p}} - V(\mathbf{p}), \mathbf{p}) = (E_{\mathbf{p}}^*, \mathbf{p})\tag{12}$$

with the energy $E_{\mathbf{p}}^* = \sqrt{\mathbf{p}^2 + M_{\mathbf{p}}^{*2}}$ expressed in terms of the effective mass $M_{\mathbf{p}}^*$, i.e., the nucleon mass modified by the scalar potential, $M_{\mathbf{p}}^* \equiv M + S(\mathbf{p})$. Analogously, an effective transferred four-momentum can be introduced:

$$Q^{*\mu} \equiv \left( P_f^{*\mu} - P_i^{*\mu} \right) = (\omega^*, \mathbf{q})\tag{13}$$

with the effective energy transferred $\omega^* = \omega - \Delta V$, and $\Delta V \equiv V(\mathbf{p}_f) - V(\mathbf{p}_i)$. Note that $Q^{*\mu}$ and $Q^\mu$ only differ in the time component. For clarity in the notation that follows, we introduce the term $\Delta V^\mu \equiv Q^\mu - Q^{*\mu}$, where only the 0 (time) component is different from zero, that is, $\Delta V^0 \equiv \Delta V = V(\mathbf{p}_f) - V(\mathbf{p}_i)$.

Finally, by performing the integral over $p_0$, the imaginary part of the polarization propagator turns out:

$$\text{Im}\,\Pi^{\mu\nu} = -\int \frac{\mathbf{p}_i^2 d|\mathbf{p}_i| d\cos\theta}{4\pi E_{\mathbf{p}_i}^* E_{\mathbf{p}_f}^*} \Theta(|\mathbf{p}_f| - k_F)\Theta(k_F - |\mathbf{p}_i|)\delta(E_{\mathbf{p}_f} - E_{\mathbf{p}_i} - q_0)T^{\mu\nu},\tag{14}$$

where $\theta$ is the angle between $\mathbf{p}_i$ and $\mathbf{q}$. The single-nucleon tensor is given by:

$$T^{\mu\nu} = \left( \slashed{P}_f^* + M_f^* \right)\widetilde{\Gamma}^\mu(\slashed{P}_i^* + M_i^*)\widetilde{\Gamma}^\nu.\tag{15}$$

In Appendix A, we present the final expression obtained for the tensor $T^{\mu\nu}$, where the nucleon form factors have been redefined and a set of dimensionless variables have been introduced to remain consistent with former publications [39].

## 2.1. Tensor Components Involved in the Weak Responses

Here we show the results obtained for the components of the single-nucleon tensor involved in the polarization propagator needed for the weak response functions. We make use of the dimensionless variables introduced in Appendix A.

**Charged-charged contribution**

$$
T_{CC} = T^{00} = \frac{\left(M_f^* + M_i^*\right)^2}{2} \left\{ -\frac{\tilde{\kappa}^{*2}}{\tilde{\tau}^*} \left[ \tilde{\tau}^* \tilde{G}_M^{*2} + G_A^2 \left(1 + \tilde{\tau}^*\right) \right] \right.
$$

$$
+ \left(\tilde{\epsilon}_i^* + \tilde{\lambda}^*\right)^2 \left[ \frac{\tilde{\tau}^* \tilde{G}_M^{*2} + \tilde{G}_E^{*2}}{\left(1 + \tilde{\tau}^*\right)} + G_A^2 \right] + \frac{\tilde{\lambda}^{*2}}{\tilde{\tau}^*} \left( G_A - \tilde{\tau}^* \tilde{G}_P^* \right)^2
$$

$$
+ 2 \frac{\left(M_f^* - M_i^*\right)}{\left(M_f^* + M_i^*\right)} \left(\tilde{\lambda}^{*2} + \tilde{\epsilon}^* \tilde{\lambda}^*\right) \left[ \frac{\tilde{G}_M^* \left(\tilde{G}_M^* - \tilde{G}_E^*\right)}{\left(1 + \tilde{\tau}^*\right)} + G_A \tilde{G}_P^* \right]
$$

$$
- \frac{\left(M_f^* - M_i^*\right)^2}{\left(M_f^* + M_i^*\right)^2} \left[ \tilde{G}_M^{*2} - \tilde{\lambda}^{*2} \tilde{G}_P^{*2} \right]
$$

$$
+ \Delta \tilde{V}^* \left[ \left( 2 \frac{\left(M_f^* - M_i^*\right)}{M_f^* + M_i^*} \tilde{\epsilon}_i^* - 4\tilde{\lambda}^* \frac{M_i^*}{M_f^* + M_i^*} \right) G_A \tilde{G}_P^* \right.
$$

$$
\left. \left. + \left(2\lambda^* + \Delta \tilde{V}^*\right) \tilde{G}_P^{*2} \left( \frac{\left(M_f^* - M_i^*\right)^2}{\left(M_f^* + M_i^*\right)^2} + \tilde{\tau}^* \right) \right] \right\}. \tag{16}
$$

**Longitudinal contribution**

$$
T_{LL} = T^{33} = \frac{\left(M_f^* + M_i^*\right)^2}{2} \left\{ -\frac{\tilde{\lambda}^{*2}}{\tilde{\tau}^*} \left[ \tilde{\tau}^* \tilde{G}_M^{*2} + G_A^2 \left(1 + \tilde{\tau}^*\right) \right] \right.
$$

$$
+ \frac{\tilde{\lambda}^{*2}}{\tilde{\kappa}^{*2}} \left[ \left(\tilde{\epsilon}_i^* + \tilde{\lambda}^*\right)^2 + \frac{\Delta \tilde{m}^{*2}}{\tilde{\lambda}^{*2}} \left( \Delta \tilde{m}^{*2} - 2\tilde{\lambda}^* (\tilde{\epsilon}_i^* + \tilde{\lambda}^*) \right) \right] \left[ \frac{\tilde{\tau}^* G_M^{*2} + \tilde{G}_E^{*2}}{\left(1 + \tilde{\tau}^*\right)} + G_A^2 \right]
$$

$$
+ \frac{\tilde{\kappa}^{*2}}{\tilde{\tau}^*} \left( G_A - \tilde{\tau}^* \tilde{G}_p^* \right)^2 + \frac{\left(M_f^* - M_i^*\right)^2}{\left(M_f^* + M_i^*\right)^2} \left[ \tilde{G}_M^{*2} + \tilde{\kappa}^{*2} \tilde{G}_P^{*2} \right]
$$

$$
+ 2 \frac{\left(M_f^* - M_i^*\right)}{\left(M_f^* + M_i^*\right)} \left[ \tilde{\lambda}^* (\tilde{\epsilon}_i^* + \tilde{\lambda}^*) - \Delta \tilde{m}^{*2} \right] \left[ \frac{\tilde{G}_M^* \left(\tilde{G}_M^* - \tilde{G}_E^*\right)}{\left(1 + \tilde{\tau}^*\right)} + G_A \tilde{G}_P^* \right]
$$

$$
+ \Delta \tilde{V}^* \left[ \left( 2\tilde{\epsilon}^* \frac{\left(M_f^* - M_i^*\right)}{\left(M_f^* + M_i^*\right)} - 4\tilde{\lambda}^* \frac{M_i^*}{\left(M_f^* + M_i^*\right)} \right) \left[ \frac{\tilde{G}_M^* \left(\tilde{G}_M^* - \tilde{G}_E^*\right)}{\left(1 + \tilde{\tau}^*\right)} \right] \right]
$$

$$
+ \frac{2\tilde{\lambda}^*}{\tilde{\kappa}^{*2}} \left[ \tilde{\tau}^* \left(\tilde{\epsilon}_i^* + \tilde{\lambda}^*\right)^2 - \Delta \tilde{m}^{*2} \left( \Delta \tilde{m}^{*2} + \frac{\left(\tilde{\tau}^* - \tilde{\lambda}^{*2}\right)}{\tilde{\lambda}^*} (\tilde{\epsilon}_i^* + \tilde{\lambda}^*) \right) \right] \frac{\left(\tilde{G}_M^* - \tilde{G}_E^*\right)^2}{\left(1 + \tilde{\tau}^*\right)^2}
$$

$$
\left. + \frac{\Delta \tilde{V}^*}{\tilde{\kappa}^{*2}} \left[ \tilde{\tau}^* \left(\tilde{\epsilon}_i^* + \tilde{\lambda}^*\right)^2 - \tilde{\kappa}^{*2} - \Delta \tilde{m}^{*2} \left( \Delta \tilde{m}^{*2} - 2\tilde{\lambda}^* (\tilde{\epsilon}_i^* + \tilde{\lambda}^*) \right) \right] \frac{\left(\tilde{G}_M^* - \tilde{G}_E^*\right)^2}{\left(1 + \tilde{\tau}^*\right)^2} \right\}. \tag{17}
$$

**Interference charged-longitudinal contribution**

$$
T_{CL} = -\frac{T^{03} + T^{30}}{2} = -\frac{\left(M_f^* + M_i^*\right)^2}{2} \left\{ -\frac{\tilde{\kappa}^{*2}}{\tilde{\tau}^*} \left[ \frac{\tilde{\lambda}^*}{\tilde{\kappa}^*} \left( \tilde{\tau}^* \tilde{G}_M^{*2} + G_A^2 \left( 1 + \tilde{\tau}^* \right) \right) \right] \right.
$$

$$
+ \frac{\tilde{\lambda}^*}{\tilde{\kappa}^*} \left[ \left( \tilde{\epsilon}_i^* + \tilde{\lambda}^* \right)^2 - \frac{\Delta \tilde{m}^{*2}}{\tilde{\lambda}^*} \left( \tilde{\epsilon}_i^* + \tilde{\lambda}^* \right) \right] \left[ \frac{\tilde{\tau}^* \tilde{G}_M^{*2} + \tilde{G}_E^{*2}}{\left( 1 + \tilde{\tau}^* \right)} + G_A^2 \right]
$$

$$
+ \frac{\tilde{\lambda}^* \tilde{\kappa}^*}{\tilde{\tau}^*} \left( G_A - \tilde{\tau}^* \tilde{G}_P^* \right)^2 + \frac{\left( M_f^* - M_i^* \right)^2}{\left( M_f^* + M_i^* \right)^2} \tilde{\lambda}^* \tilde{\kappa}^* \tilde{G}_P^{*2}
$$

$$
+ \frac{\left( M_f^* - M_i^* \right)}{\left( M_f^* + M_i^* \right)} \frac{1}{\tilde{\kappa}^*} \left[ \left( \tilde{\epsilon}_i^* + \tilde{\lambda}^* \right) \left( \tilde{\kappa}^{*2} + \tilde{\lambda}^{*2} \right) - \Delta \tilde{m}^{*2} \tilde{\lambda}^* \right] \left[ \frac{\tilde{G}_M^* \left( \tilde{G}_M^* - \tilde{G}_E^* \right)}{\left( 1 + \tilde{\tau}^* \right)} + G_A \tilde{G}_P^* \right]
$$

$$
+ \frac{\Delta \tilde{V}^*}{\tilde{\kappa}^*} \left[ \left( \left( \tilde{\epsilon}_i^* + \tilde{\lambda}^* \right)^2 \tilde{\tau}^* + \tilde{\lambda}^* \Delta \tilde{m}^{*2} \left( \tilde{\epsilon}_i^* + \tilde{\lambda}^* \right) \right) \frac{\left( \tilde{G}_M^* - \tilde{G}_E^* \right)^2}{\left( 1 + \tilde{\tau}^* \right)^2} \right.
$$

$$
+ \tilde{\kappa}^{*2} \left[ \frac{\left( M_f^* - M_i^* \right)^2}{\left( M_f^* + M_i^* \right)} + \tilde{\tau}^* \right] \tilde{G}_P^{*2}
$$

$$
\left. + \left[ \frac{\left( M_f^* - M_i^* \right)}{\left( M_f^* + M_i^* \right)} \left( \tilde{\lambda}^* \tilde{\epsilon}_i^* - \tilde{\tau}^* - \Delta \tilde{m}^{*2} \right) - \frac{2 M_i^* \tilde{\kappa}^{*2}}{\left( M_f^* + M_i^* \right)} \right] \left[ \frac{\tilde{G}_M^* \left( \tilde{G}_M^* - \tilde{G}_E^* \right)}{\left( 1 + \tilde{\tau}^* \right)} + G_A \tilde{G}_P^* \right] \right] \right\}, \quad (18)
$$

**Transverse contribution**

$$
T_T = \left( T^{11} + T^{22} \right) = \frac{\left( M_f^* + M_i^* \right)^2}{2} \left\{ 2 \left[ \tilde{\tau}^* \tilde{G}_M^{*2} + G_A^2 \left( 1 + \tilde{\tau}^* \right) \right] \right.
$$

$$
+ \frac{1}{\tilde{\kappa}^{*2}} \left[ \tilde{\tau}^* \left( \tilde{\epsilon}_i^* + \tilde{\lambda}^* \right)^2 - \tilde{\kappa}^{*2} \left( \tilde{\tau}^* + \tilde{m}_i^{*2} \right) - \Delta \tilde{m}^{*2} \left( \Delta \tilde{m}^{*2} - 2 \left( \tilde{\lambda}^* \tilde{\epsilon}_i^* - \tilde{\tau}^* \right) \right) \right]
$$

$$
\left[ \frac{\tilde{\tau}^* \left( \tilde{G}_M^* \right)^2 + \left( \tilde{G}_E^* \right)^2}{\left( 1 + \tilde{\tau}^* \right)} + G_A^2 \right] + 2 \frac{\left( M_f^* - M_i^* \right)^2}{\left( M_f^* + M_i^* \right)^2} \tilde{G}_M^{*2} \right\}
$$

$$
- \Delta \tilde{V}^* \left[ \frac{4}{\left( M_f^* + M_i^* \right)} \left( 2 \tilde{\lambda}^* M_i^* - \tilde{\epsilon}_i^* \left( M_f^* - M_i^* \right) \right) \right] \left[ \frac{\tilde{G}_M^* \left( \tilde{G}_M^* - \tilde{G}_E^* \right)}{\left( 1 + \tilde{\tau}^* \right)} \right]
$$

$$
- 2 \Delta \tilde{V}^* \left[ \left( \tilde{\epsilon}_i^* + \tilde{\lambda}^* \right)^2 - \left( \tilde{\kappa}^{*2} + 1 \right) \right] \frac{\left( \tilde{G}_M^* - \tilde{G}_E^* \right)^2}{\left( 1 + \tilde{\tau}^* \right)^2} + \frac{1}{2 \tilde{\kappa}^{*2}} \left( 2 \tilde{\lambda}^* + \Delta \tilde{V}^* \right)
$$

$$
\left. \left[ \tilde{\tau}^* \left( \tilde{\epsilon}_i^* + \tilde{\lambda}^* \right)^2 - \tilde{\kappa}^{*2} \left( \tilde{\tau}^* + \tilde{m}_i^{*2} \right) - \Delta \tilde{m}^{*2} \left( \Delta \tilde{m}^{*2} - 2 \left( \tilde{\lambda}^* \tilde{\epsilon}_i^* - \tilde{\tau}^* \right) \right) \right] \frac{\left( \tilde{G}_M^* - \tilde{G}_E^* \right)^2}{\left( 1 + \tilde{\tau}^* \right)^2} \right\}. \quad (19)
$$

**Axial-vector transverse contribution**

$$T_{T'} = -\frac{i}{2}\left(T^{12} - T^{21}\right)$$

$$= -\frac{\left(M_f^* + M_i^*\right)^2}{\tilde{\kappa}^*}\left[\tilde{\tau}^*\left(\tilde{\epsilon}_i^* + \tilde{\lambda}^*\right) + \tilde{\lambda}^*\Delta\tilde{m}^{*2} - \Delta\tilde{V}^*\left(\tilde{\lambda}^*\tilde{\epsilon}_i^* - \tilde{\tau}^* - \Delta\tilde{m}^{*2}\right)\right]G_A\tilde{G}_M^*. \quad (20)$$

In all the previous expressions, we have introduced $\Delta\tilde{V}^* \equiv \frac{\Delta V}{M_f^* + M_i^*}$, $\tilde{m}_{i,f}^* \equiv \frac{2M_{i,f}^*}{M_i^* + M_f^*}$, and $\Delta\tilde{m}^{*2} \equiv \frac{\Delta m^{*2}}{(M_i^* + M_f^*)^2}$.

*2.2. Weak Nuclear Responses*

Starting from the previous expressions, the different components of the polarization tensor can be computed by performing the integrals numerically. The resulting response functions will be shown in the next section. Here we consider the phenomenological scalar and vector potentials, $S(\mathbf{p})$, $V(\mathbf{p})$, adjusted to polynomials in $\mathbf{p}$. We use the same expressions already considered for electron scattering (see [39]) and taken from [44,45], that is:

$$S(\mathbf{p}) = \alpha S_0\left[a_0 + a_1 T(\mathbf{p}) + a_2 T(\mathbf{p})^2\right]$$

$$V(\mathbf{p}) = \alpha V_0\left[b_0 + b_1 T(\mathbf{p}) + b_2 T(\mathbf{p})^2\right]$$

with $S_0 = -0.431$ GeV and $V_0 = 0.354$ GeV for the constant scalar and vector potentials. The term $T(\mathbf{p})$ is the kinetic energy of the nucleon, and the parameters $a_i, b_i$ are given by: $a_0 = 0.97$, $a_1 = -0.66$, $a_2 = 0.28$, $b_0 = 0.97$, $b_1 = -0.97$, and $b_2 = 0.33$. The factor $\alpha = \left(k_F/k_F^0\right)^3$ represents an average over the nuclear volume, with $k_F^0 = 0.257$ GeV/c the standard value of the Fermi momentum for nuclear matter.

Hereinafter we focus on some particular cases where the integrals can be solved analytically, thus providing explicit expressions for the polarization tensor, and likewise for the nuclear response functions. In the case of vector potentials to be almost equal in the initial and final states, i.e., $V(\mathbf{p}_i) \simeq V(\mathbf{p}_f)$, the term $\Delta V$ tends to zero, and the effective energy transfer, $\omega^*$, corresponds to $\omega$. Thus the tensor does not depend on $V$, and the only dependence on the scalar potential, $S$, appears on the effective masses: $M_{i,f}^*$. Moreover, assuming equal scalar potentials in the initial and final states, i.e., $M_i^* = M_f^* \equiv M^*$, the expression for the single-nucleon tensor reduces to:

$$\begin{aligned}
T^{\mu\nu} &= -2M^{*2}\left(g^{\mu\nu} - \frac{Q^{*\mu}Q^{*\nu}}{Q^{*2}}\right)\left[\tilde{\tau}^*\tilde{G}_M^{*2} + G_A^2\left(1 + \tilde{\tau}^*\right)\right] \\
&+ \left(2P_i^{*\mu}P_i^{*\nu} + P_i^{*\mu}Q^{*\nu} + Q^{*\mu}P_i^{*\nu} + \frac{Q^{*\mu}Q^{*\nu}}{2}\right)\left[\frac{\tilde{\tau}^*\tilde{G}_M^{*2} + \tilde{G}_E^{*2}}{\left(1 + \tilde{\tau}^*\right)} + G_A^2\right] \\
&+ \frac{Q^{*\mu}Q^{*\nu}}{2\tilde{\tau}^*}\left(G_A - \tilde{\tau}^*\tilde{G}_P^*\right)^2 - 2i\epsilon^{\mu\nu\beta\lambda}P_{i\beta}^*Q_\lambda^*G_A\tilde{G}_M^*.
\end{aligned} \quad (21)$$

The different contributions, *CC*, *LL* and *CL*, that enter in the longitudinal channel are given by:

$$T_{CC} = 2M^{*2} \left\{ -\frac{\tilde{\kappa}^{*2}}{\tilde{\tau}^*} \left[ \tilde{\tau}^* \left( \tilde{G}_M^* \right)^2 + \left( G_A \right)^2 \left( 1 + \tilde{\tau}^* \right) \right] \right.$$
$$\left. + (\tilde{\epsilon}_i^* + \tilde{\lambda}^*)^2 \left[ \frac{\tilde{\tau}^* \left( \tilde{G}_M^* \right)^2 + \left( \tilde{G}_E^* \right)^2}{\left( 1 + \tilde{\tau}^* \right)} + \left( G_A \right)^2 \right] + \frac{\tilde{\lambda}^{*2}}{\tilde{\tau}^*} \left( G_A - \tilde{\tau}^* \tilde{G}^* \right)^2 \right\}, \quad (22)$$

$$T_{CL} = 2M^{*2} \left\{ \frac{\tilde{\kappa}^{*2}}{\tilde{\tau}^*} \left[ \frac{\tilde{\lambda}^*}{\tilde{\kappa}^*} \left( \tilde{\tau}^* \left( \tilde{G}_M^* \right)^2 + \left( G_A \right)^2 \left( 1 + \tilde{\tau}^* \right) \right) \right] \right.$$
$$\left. - \frac{\tilde{\lambda}^*}{\tilde{\kappa}^*} \left[ (\tilde{\epsilon}_i^* + \tilde{\lambda}^*)^2 - \frac{\Delta \tilde{m}^{*2}}{\tilde{\lambda}^*} (\tilde{\epsilon}_i^* + \tilde{\lambda}^*) \right] \left[ \frac{\tilde{\tau}^* \left( \tilde{G}_M^* \right)^2 + \left( \tilde{G}_E^* \right)^2}{\left( 1 + \tilde{\tau}^* \right)} + \left( G_A \right)^2 \right] \right.$$
$$\left. - \frac{\tilde{\lambda}^* \tilde{\kappa}^*}{\tilde{\tau}^*} \left( G_A - \tilde{\tau}^* \tilde{G}_P^* \right)^2 \right\}, \quad (23)$$

$$T_{LL} = 2M^{*2} \left\{ -\frac{\tilde{\lambda}^{*2}}{\tilde{\tau}^*} \left[ \tilde{\tau}^* \left( \tilde{G}_M^* \right)^2 + \left( G_A \right)^2 \left( 1 + \tilde{\tau}^* \right) \right] \right.$$
$$\left. + \frac{\tilde{\lambda}^{*2}}{\tilde{\kappa}^{*2}} \left[ (\tilde{\epsilon}_i^* + \tilde{\lambda}^*)^2 \left[ \frac{\tilde{\tau}^* \left( G_M^* \right)^2 + \left( \tilde{G}_E^* \right)^2}{\left( 1 + \tilde{\tau}^* \right)} + \left( G_A \right)^2 \right] \right] \right.$$
$$\left. + \frac{\tilde{\kappa}^{*2}}{\tilde{\tau}^*} \left( G_A - \tilde{\tau}^* \tilde{G}_p^* \right)^2 \right\}, \quad (24)$$

while the transverse, *T*, and the axial-vector transverse, *T'*, contributions read:

$$T_T = 2M^{*2} \left\{ 2 \left[ \tilde{\tau}^* \tilde{G}_M^{*2} + G_A^2 \left( 1 + \tilde{\tau}^* \right) \right] \right.$$
$$\left. + \frac{1}{\tilde{\kappa}^{*2}} \left[ \tilde{\tau}^* \left( \tilde{\epsilon}_i^* + \tilde{\lambda}^* \right)^2 - \tilde{\kappa}^{*2} \left( \tilde{\tau}^* + \tilde{m}_i^{*2} \right) \right] \left[ \frac{\tilde{\tau}^* \tilde{G}_M^{*2} + \tilde{G}_E^{*2}}{\left( 1 + \tilde{\tau}^* \right)} + G_A^2 \right] \right\}, \quad (25)$$

$$T_{T'} = -4M^{*2} \left[ \frac{\tilde{\tau}^*}{\tilde{\kappa}^*} \left( \tilde{\epsilon}_i^* + \tilde{\lambda}^* \right) G_A \tilde{G}_M^* \right]. \quad (26)$$

Finally, in the simple case of constant scalar and vector potentials (and equal for the initial and final states), referred to as *CtSV*, the integrals in the polarization propagator can be solved analytically, and explicit expressions for the weak nuclear response functions emerge:

$$R_{CtSV}^{CC} = \frac{3\mathcal{N}}{4M^* \tilde{\kappa}^* \tilde{\eta}_F^{*3}} \Theta (\tilde{\epsilon}_F^* - \tilde{\Gamma}^*)(\tilde{\epsilon}_F^* - \tilde{\Gamma}^*)$$
$$\left\{ \frac{\tilde{\kappa}^{*2}}{\tilde{\tau}^*} \left[ \left( \tilde{G}_E^{*2} + \tilde{\Delta}^* \frac{\tilde{\tau}^* \tilde{G}_M^{*2} + \tilde{G}_E^{*2}}{(1 + \tilde{\tau}^*)} \right) + \tilde{\Delta}^* G_A^2 \right] + \frac{\tilde{\lambda}^{*2}}{\tilde{\tau}^*} \left( G_A - \tilde{\tau}^* \tilde{G}_P^* \right)^2 \right\} \quad (27)$$

$$R_{CtSV}^{CL} = \frac{3\mathcal{N}}{4M^*\tilde{\kappa}^*\tilde{\eta}_F^{*3}}\Theta(\tilde{\epsilon}_F^* - \tilde{\Gamma}^*)(\tilde{\epsilon}_F^* - \tilde{\Gamma}^*)$$

$$\left\{\frac{\tilde{\lambda}^*}{\tilde{\kappa}^*}\left[\frac{\tilde{\kappa}^{*2}}{\tilde{\tau}^*}\left(\tilde{G}_E^{*2} + \tilde{\Delta}^*\frac{\tilde{\tau}^*\tilde{G}_M^{*2} + \tilde{G}_E^{*2}}{(1+\tilde{\tau}^*)}\right) + \frac{\tilde{\kappa}^{*2}}{\tilde{\tau}^*}\tilde{\Delta}^* G_A^2\right] + \frac{\tilde{\lambda}^*\tilde{\kappa}^*}{\tilde{\tau}^*}\left(G_A - \tilde{\tau}^*\tilde{G}_P^*\right)^2\right\}$$

$$(28)$$

$$R_{CtSV}^{LL} = \frac{3\mathcal{N}}{4M^*\tilde{\kappa}^*\tilde{\eta}_F^{*3}}\Theta(\tilde{\epsilon}_F^* - \tilde{\Gamma}^*)(\tilde{\epsilon}_F^* - \tilde{\Gamma}^*)$$

$$\left\{\frac{\tilde{\lambda}^{*2}}{\tilde{\kappa}^{*2}}\left[\frac{\tilde{\kappa}^{*2}}{\tilde{\tau}^*}\left(\tilde{G}_E^{*2} + \tilde{\Delta}^*\frac{\tilde{\tau}^*\tilde{G}_M^{*2} + \tilde{G}_E^{*2}}{(1+\tilde{\tau}^*)}\right) + \frac{\tilde{\kappa}^{*2}}{\tilde{\tau}^*}\tilde{\Delta}^* G_A^2\right] + \frac{\tilde{\kappa}^{*2}}{\tilde{\tau}^*}\left(G_A - \tilde{\tau}^*\tilde{G}_P^*\right)^2\right\}$$

$$(29)$$

$$R_{CtSV}^{T} = \frac{3\mathcal{N}}{4M^*\tilde{\kappa}^*\tilde{\eta}_F^{*3}}\Theta(\tilde{\epsilon}_F^* - \tilde{\Gamma}^*)(\tilde{\epsilon}_F^* - \tilde{\Gamma}^*)$$

$$\left\{2\tilde{\tau}^*\tilde{G}_M^{*2} + \tilde{\Delta}^*\frac{\left(\tilde{\tau}^*\tilde{G}_M^{*2} + \tilde{G}_E^{*2}\right)}{(1+\tilde{\tau}^*)} + \left(2(1+\tilde{\tau}^*) + \tilde{\Delta}^*\right)G_A^2\right\}$$

$$(30)$$

$$R_{CtSV}^{T'} = \frac{3\mathcal{N}}{4M^*\tilde{\kappa}^*\tilde{\eta}_F^{*3}}\Theta(\tilde{\epsilon}_F^* - \tilde{\Gamma}^*)(\tilde{\epsilon}_F^* - \tilde{\Gamma}^*)\left\{\frac{\tilde{\tau}^*}{\tilde{\kappa}^*}\left[\frac{1}{2}(\tilde{\epsilon}_F^* + \tilde{\Gamma}^*) + \tilde{\lambda}^*\right]G_A\tilde{G}_M^*\right\}, \quad (31)$$

where the dimensionless Fermi energy, $\tilde{\epsilon}_F^*$, has been introduced by analogy with the definitions given in Appendix A, but replacing the nucleon energy, $E_\mathbf{p}^*$, with the Fermi one: $E_F^* = \sqrt{k_F^2 + M_{k_F}^{*2}}$ with $M_{k_F}^* = M + S(k_F)$. In the above expressions, we make use of the usual terms [21,39]:

$$\tilde{\Gamma}^* = Max\left\{\tilde{\epsilon}_F^* - 2\tilde{\lambda}^*; -\tilde{\lambda}^*\tilde{\rho}^* + \tilde{\kappa}\sqrt{\tilde{\rho}^{*2} + \frac{\tilde{m}_i^{*2}}{\tilde{\tau}^*}}\right\}, \quad (32)$$

$$\tilde{\Delta}^* = \frac{\tilde{\tau}^*}{\tilde{\kappa}^{*2}}\left[\frac{1}{3}\left(\tilde{\epsilon}_F^{*2} + \tilde{\epsilon}^*\tilde{\Gamma}^* + \tilde{\Gamma}^{*2}\right) + \tilde{\lambda}^*\left(\tilde{\epsilon}^* + \tilde{\Gamma}^*\right) + \tilde{\lambda}^{*2}\right] - \left(\tilde{\tau}^* + 1\right). \quad (33)$$

A detailed analysis of the general function, $\Gamma$, and its specific behavior at different kinematical situations is presented in [22]. In Equation (32), we have introduced the notation $\tilde{\rho}^*$ by analogy with the expression for $\rho^*$ presented in Appendix A, that is, $\tilde{\rho}^* = \left(1 + \frac{\Delta\tilde{m}^{*2}}{\tilde{\tau}^*}\right)$.

The study of scaling and superscaling in inclusive quasielastic (QE) electron scattering reactions has been presented in detail in previous works [8,22–27]. Its extension to inclusive charged-current neutrino processes can be reviewed in [21]. From these studies, not only is the scaling also fulfilled for neutrino reactions, that is, the independence of the momentum transfer $q$ (scaling of the first kind) and of the nuclear system (scaling of the second kind), but also the scaling function is similar to the one obtained from electron scattering. In this work, we provide a systematic analysis of the scaling/superscaling phenomenon within the framework of the relativistic mean field approach in nuclear matter. Following our previous work on electron scattering, here we extend our investigation to neutrino processes, analyzing the effects introduced by the use of scalar and vector potentials considering both momentum-independent and momentum-dependent potentials.

Our main interest is centered on the scaling behavior at high transferred momentum, where $\tilde{\Gamma}^*$ is determined by the second choice in (32). The response functions in (27)–(31)

can also be expressed by introducing a general dimensionless scaling variable, $\tilde{\psi}^*$, and the superscaling function, $f(\tilde{\psi}^*)$, that are given by (see [39] for details):

$$
\tilde{\psi}^* = \frac{1}{\sqrt{\tilde{\xi}_F^*}} \frac{\tilde{\lambda}^* \tilde{m}_i^* - \tilde{\tau}^* \tilde{\rho}^*}{\sqrt{\tilde{\tau}^*(\tilde{\lambda}^* \tilde{\rho}^* + \tilde{m}_i^*) + \tilde{\kappa}^* \sqrt{\tilde{\tau}^*\left(\tilde{\tau}^* \tilde{\rho}^{*2} + \tilde{m}^{*2}\right)}}} \,, \tag{34}
$$

$$
f(\tilde{\psi}^*) = \frac{3}{4}\left(1 - \tilde{\psi}^{*2}\right)\Theta\left(1 - \tilde{\psi}^{*2}\right) \tag{35}
$$

with $\tilde{\xi}_F^* = \left(\tilde{\epsilon}_F^* - \tilde{m}_F^*\right)$. Notice that, in the absence of scalar potentials, $S = 0$, the usual expression for the scaling variable as given in previous references [24,25] is recovered.

For completeness, in this study we also consider the case of the Relativistic Plane Wave Impulse Approximation (RPWIA), that is, no scalar vector potentials in the final state. On the contrary, the initial nucleon states are described by Dirac wave function solutions in the presence of $S(\mathbf{p}_i)$ and $V(\mathbf{p}_i)$. In the next section, we present a detailed study of the weak nuclear responses and scaling functions corresponding to the different approaches considered in this work.

## 3. Discussion of Results

In this section, we present the results obtained for the nuclear weak charged-current responses and scaling functions corresponding to three nuclear systems $^{12}$C, $^{16}$O, and $^{40}$Ca, using the different approaches discussed in previous sections for the description of the nuclear dynamics. We also show the pure RFG predictions as reference. The results cover a kinematical region from low–intermediate momentum transferred, $q = 0.5$ GeV/c, to moderate–high values, $q = 1.5$ GeV/c.

### 3.1. Nuclear Responses

In Figures 1 and 2, we present the response functions for $^{12}$C corresponding to the longitudinal (*CC*, *CL*, and *LL*) and transverse (*T* and *T*′) channels, respectively. The value of the Fermi momentum has been set to 228 MeV/c. The panels show the different nuclear responses in terms of the energy transferred, $\omega$, for fixed values of $q$: 0.5 GeV/c (top panels), 1 GeV/c (mid), and 1.5 GeV/c (bottom). In all the situations, we perform a comparison between the predictions given by RFG (black solid line), RPWIA (green dot-dashed), constant scalar and vector potentials in both the initial and final nucleon states, denoted as CtSV (blue dotted), and energy-dependent scalar and vector potentials, defined as EDSV (red dashed).

We start the discussion with the three responses involved in the longitudinal channel (Figure 1). As observed, the scalar and vector potentials involved produce significant differences between the various models that also depend on the particular response analyzed and the kinematics selected. Regarding the EDSV and CtSV models, their predictions are not remarkably different at $q = 0.5$ GeV/c, but they are shifted to larger $\omega$ values compared to the RFG response, also getting higher values for the maximum and more asymmetry. This is particularly true for the *CL* and *LL* responses, where the discrepancy with RFG can differ by more than a factor of 2 (*LL* response). This behavior is clearly in contrast with the results obtained for the pure electromagnetic longitudinal response [39], where the RFG magnitude at the maximum was similar to the CtSV and EDSV predictions.

It is also worth mentioning that in the relativistic plane wave impulse approximation (RPWIA), the minimum value of $\omega$ allowed by the kinematics is significantly higher than the one for the other models. This is connected with the role that potentials play only in the initial state. On the contrary, the maximum $\omega$ value allowed is located between the pure RFG result and the CtSV/EDSV ones, where both initial and final-state potentials are considered. Concerning the overall magnitude of the RPWIA response, it is closer to the RFG one (although visible differences appear in the *CL* and *LL* channels) and significantly smaller than the EDSV and CtSV predictions.

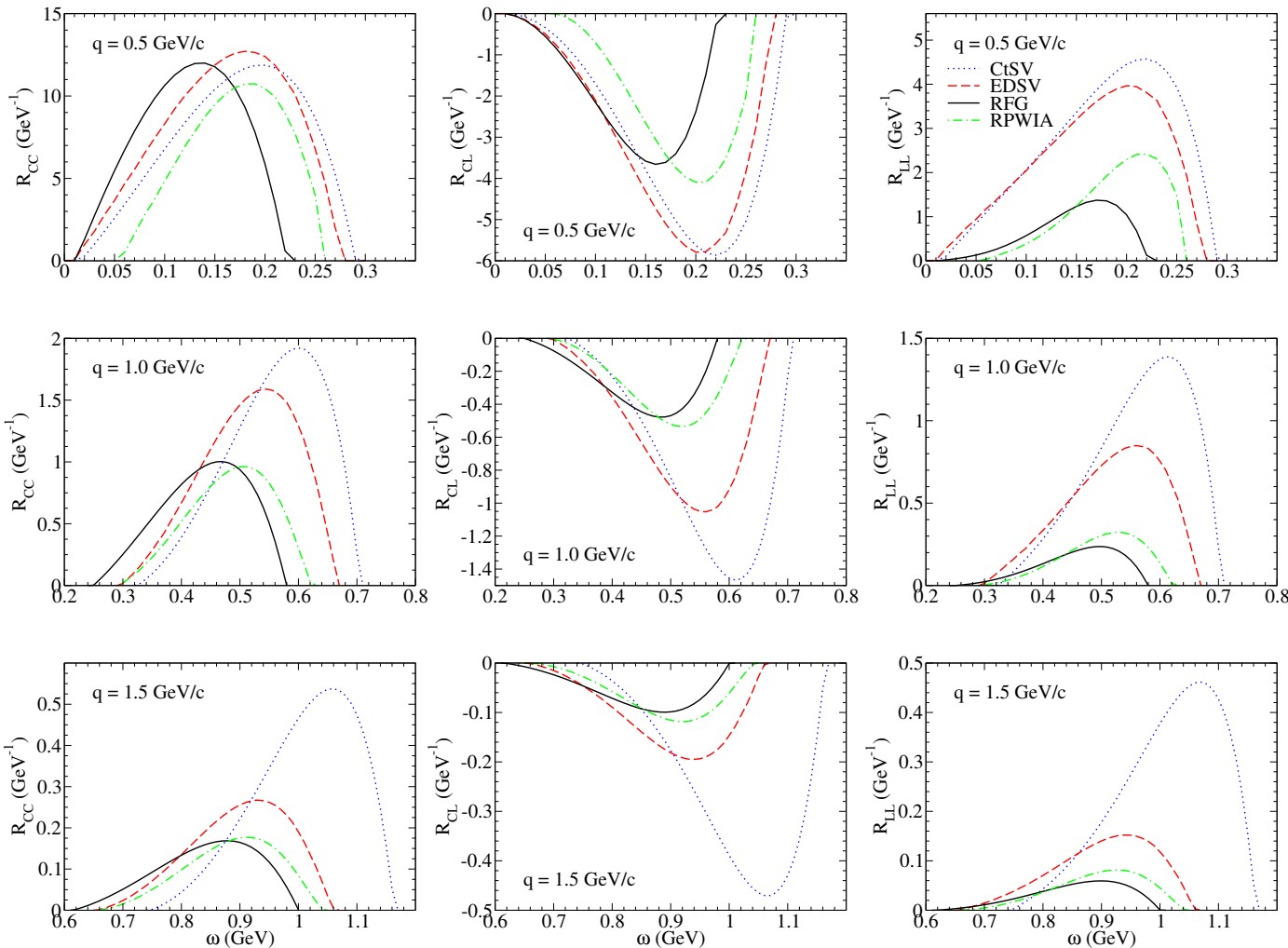

**Figure 1.** $^{12}$C weak response functions for the CC (**left** panels), CL (**mid**), and LL (**right**) channels versus the energy transfer, $\omega$. Results are shown for different values of the momentum transfer, $q$, and the models considered in the work (see text for details): RFG (black solid), EDSV (red dashed), CtSV (blue dotted), and RPWIA (green dot-dashed).

For higher $q$-values (mid and bottom panels), the results that depart the most correspond to the CtSV model (blue dotted line). This discrepancy gets larger as $q$ increases, and this is connected with the very strong energy-independent potentials involved in CtSV. As the transfer momentum, $q$, increases, one expects the effects of final state interactions (FSI) to be weaker. This is supported by the general accordance between RFG and RPWIA results, where in the latter, only energy-dependent potentials are involved in the initial state. In contrast, the more realistic EDSV calculation gives rise to larger (absolute values) responses but is still far from the CtSV predictions. Note that as the final nucleon momenta increase, the magnitude of the scalar and vector potentials diminishes. Summarizing, we observe that as the value of $q$ increases, the relative discrepancy between RPWIA and EDSV predictions gets weaker, but the CtSV model produces much larger responses due to the very strong scalar and vector potentials considered in the final state. This is in accord with the general analysis presented for the pure electromagnetic responses [39], although with significant relative discrepancies due to the axial term present in the weak sector.

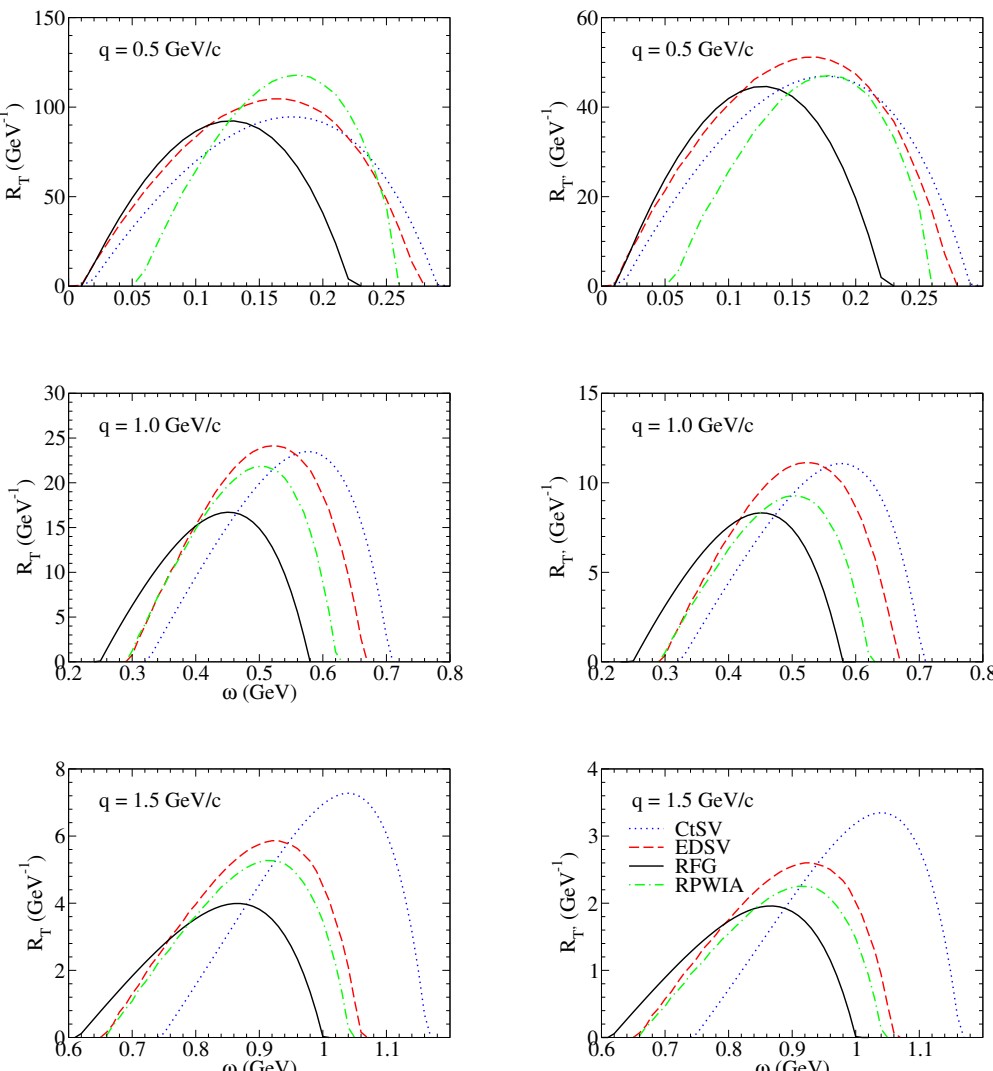

**Figure 2.** Same as Figure 1 but for the transverse and interference vector-axial transverse weak response functions, T (**left** panels) and T' (**right** panels), respectively.

The previous discussion also applies to the two transverse responses (Figure 2), but with important singularities to be noted. First, the contribution of the longitudinal response in (anti)neutrino processes is, in most kinematicial situations, negligible compared with the transverse ones. Hence, the discrepancies observed in the $T$ and $T'$ channels between the predictions of the different models will emerge in the differential cross section. As observed, the CtSV model departs the most as $q$ becomes larger. On the contrary, EDSV and RPWIA results get closer together, which is in accordance with the kinematics analyzed, where much larger values of the transferred energy and momentum (and also for the ejected nucleon momentum/energy) are involved. This implies that the strength of the potentials in the final state becomes weaker as $q$ increases. Finally, compared to the RFG prediction, the EDSV responses (likewise RPWIA) are significantly larger (particularly in the $T$ channel). This is in contrast to the results obtained for the pure electromagnetic transverse response [39], where the discrepancy between RFG and EDSV/RPWIA was much smaller. This can be connected with the axial–axial contribution in the weak responses that is absent in the electromagnetic interaction.

In Figures 3 and 4, we present the weak responses obtained with the different models for $^{16}O$ and $^{40}Ca$, respectively. The values considered for the Fermi momentum are $k_F = 230$ MeV/c ($^{16}O$) and 241 MeV/c ($^{40}Ca$). These, and the one for carbon, correspond to

the values given in [24] that were determined from a careful analysis of quasielastic $(e, e')$ data. Note that in the RFG model, the parameters that define the nuclear dependence are mostly the Fermi momentum and the energy shift, which can be taken as an average of the nucleon's binding energy [24]. As can be noticed, the curves show a similar behavior to the ones for $^{12}$C, apart from the magnitude of the responses.

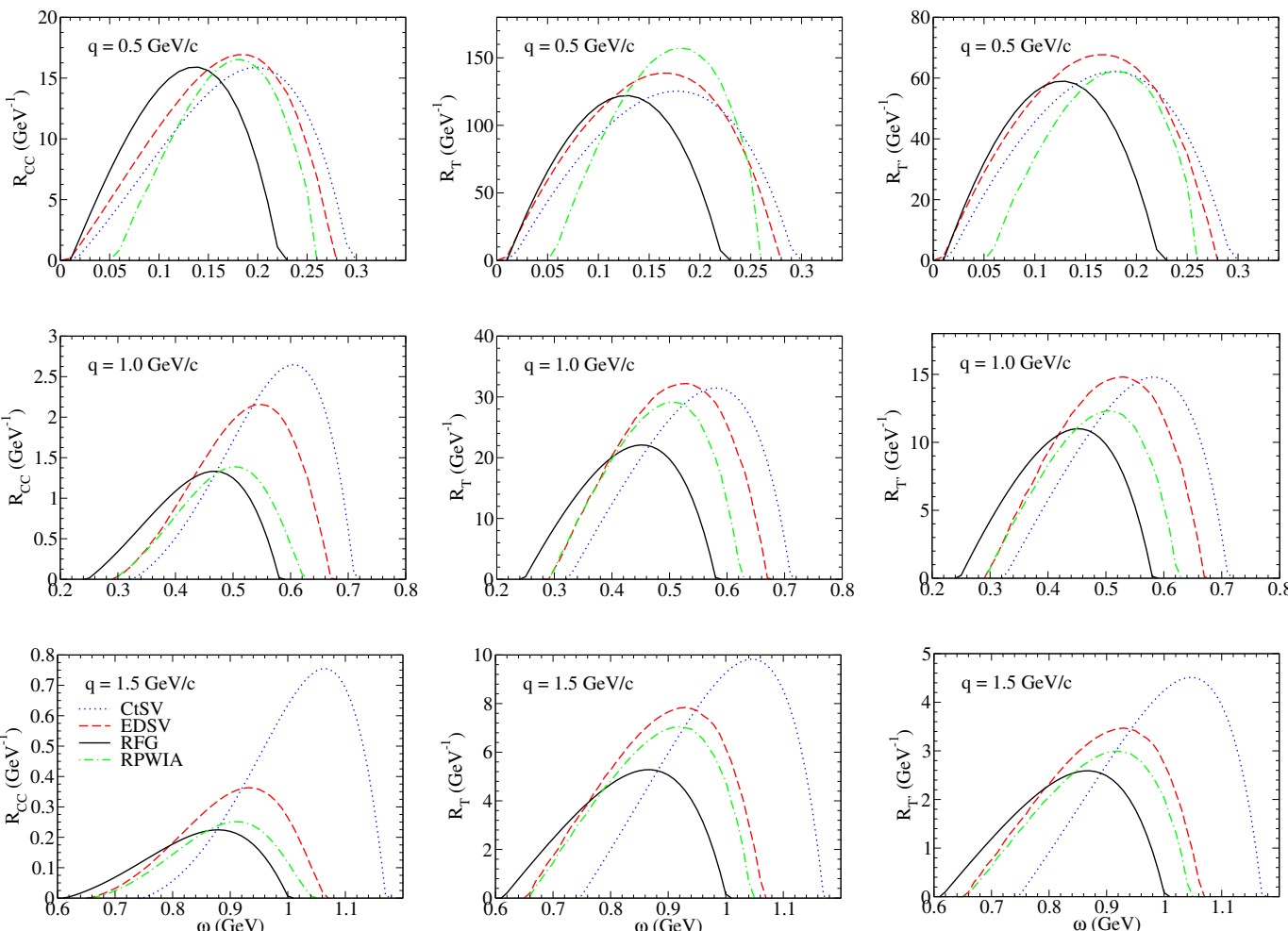

**Figure 3.** Same as Figures 1 and 2 but considering $^{16}$O as nuclear target. Note that, for the longitudinal response functions, only the CC one is included as the most representative case.

Here, for simplicity, we only show the pure charge–charge (CC) contribution to the longitudinal channel (left panels). The main deviation at high $q$ is associated with the CtSV model because of the strong scalar and vector constant potentials considered in both the initial and final states. Furthermore, whereas the various models clearly differ in the CC channel, the EDSV and RPWIA predictions for the two transverse responses do not differ too much, although their maxima clearly exceed the RFG response, particularly at higher $q$. As already discussed, these discrepancies are remarkably smaller for the pure electromagnetic responses (see [39]). Finally, it is worth noticing the behavior shown by the RPWIA predictions for calcium at $q = 0.5$ GeV/c. As noted in [39], this is a consequence of the assumptions implied by RPWIA that lead to the value of $\tilde{\tau}^*$ becoming negative.

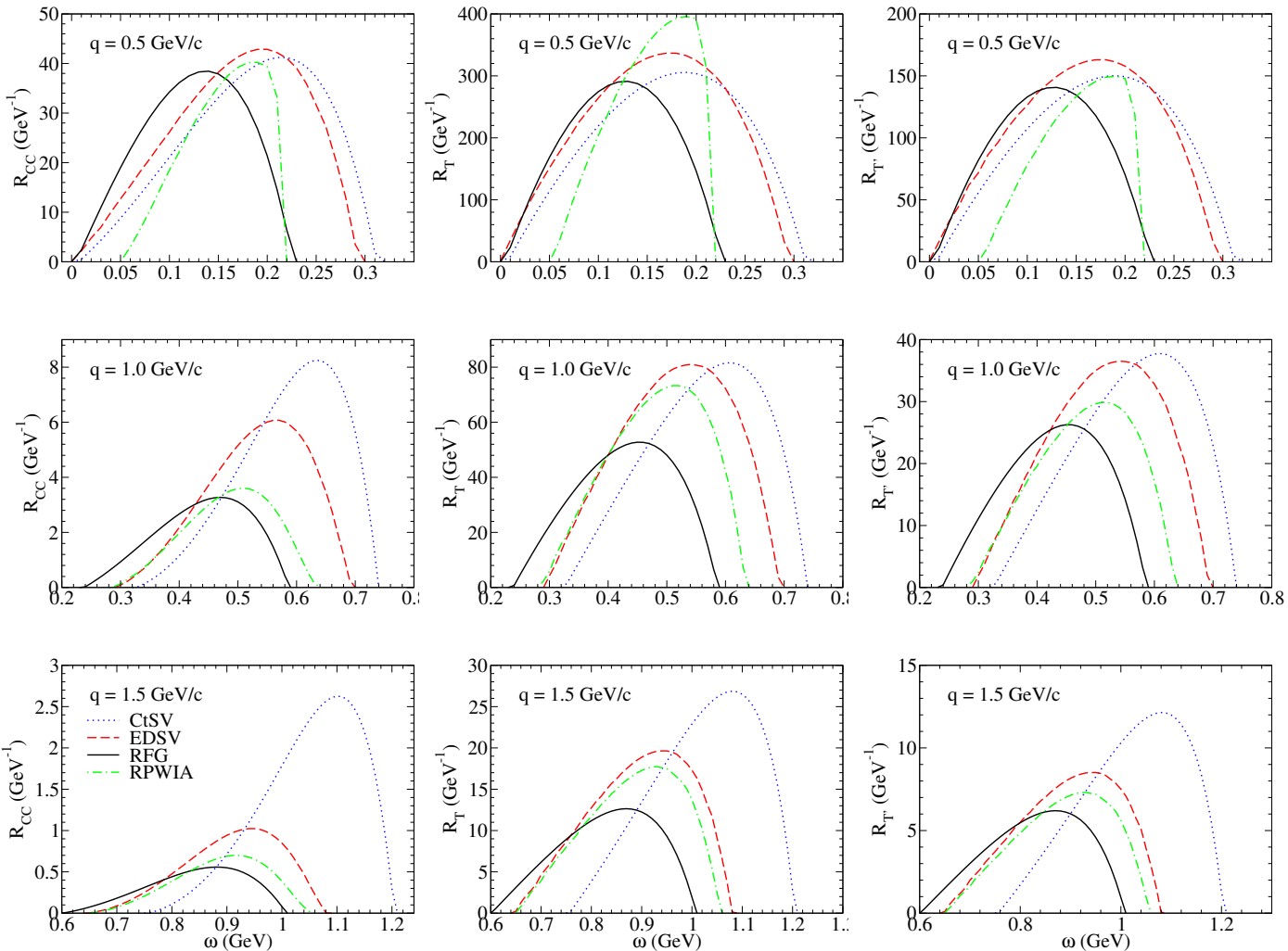

**Figure 4.** Same as Figure 3 but for $^{40}$Ca as nuclear target.

### 3.2. Scaling Functions

The analysis of the scaling and superscaling behaviour in the weak responses is shown in Figures 5–7, where we present the results for the different theoretical approaches, nuclear targets, and $q$-values, as shown in the previous section. For simplicity, we restrict our study to the transverse ($T$) and interference axial-vector transverse ($T'$) channels as the longitudinal contribution in the weak sector is very small, almost negligible, in most kinematical situations [46].

In Figure 5, we analyze the scaling of the first kind, i.e., independence of the scaling function with the momentum transfer, $q$. We have selected the EDSV and RPWIA models and show results only for $^{12}$C. In addition, for reference we present the RFG function (solid black line), i.e., $(3/4)(1 - \psi^2)\Theta(1 - \psi^2)$. The left panels correspond to the EDSV approach, that is, results obtained in the presence of energy-dependent scalar and vector potentials in the initial and final states, while the right panels refer to RPWIA. We show separately the $T$ (top panels) and $T'$ (bottom panels) scaling functions comparing the predictions corresponding to three different values of the momentum transfer: $q = 0.5$ GeV/c (brown double-dotted dashed), 1 GeV/c (red dashed), and 1.5 GeV/c (blue dot-dashed).

As observed, the EDSV and the RPWIA scaling functions, compared to the RFG, are shifted to larger values of the scaling variable, that is, larger transfer energy, $\omega$. Furthermore, the overall magnitude (value at the maximum) of both $f_T$ and $f_{T'}$ for the EDSV model is significantly larger than the RFG. A similar comment also applies to RPWIA in the $T$

channel, whereas for the axial-vector transverse response, $T'$, the RPWIA predictions are close to RFG, except for the shift in $\psi$.

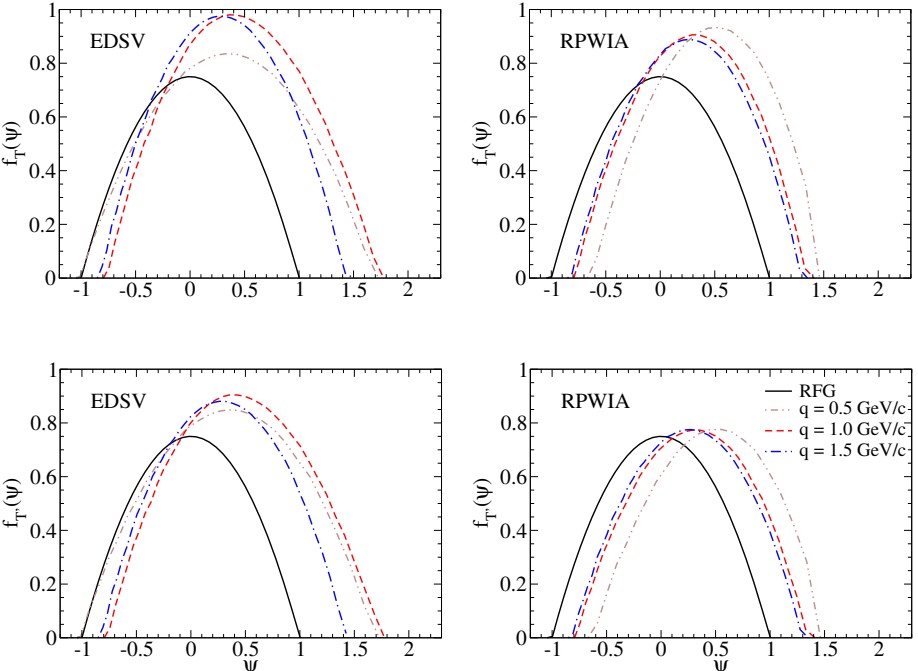

**Figure 5.** Analysis of 1st kind scaling for the transverse (**top** panels) and interference vector-axial transverse (**bottom**) $^{12}$C scaling functions at different $q$ values for the EDSV (**left** panels) and RPWIA (**right**) models. The RFG scaling function is shown in all panels as reference.

The RPWIA results in Figure 5 show that scaling of the first kind is clearly fulfilled for $q \geq 1$ GeV/c. Only the case of $q = 0.5$ GeV/c differs from the others, shifted to larger $\psi$-values. This coincides with the results for the RPWIA electromagnetic scaling functions [39]. Concerning the EDSV model, its predictions for $f_T$ and $f_{T'}$ exceed the RPWIA ones, and are much larger than the RFG. This is particularly true for the $T'$ response, and it clearly differs from the analysis in the pure electromagnetic transverse response [39], where the EDSV scaling functions (maxima) were similar to RFG. Furthermore, within the EDSV model, scaling of the first kind is better reproduced by $f_{T'}$, while it is clearly broken by $f_T$ (note the significant discrepancy at $q = 0.5$ GeV/c). We conclude that scaling behavior works better for RPWIA in the two transverse responses.

In Figure 6, we analyze the scaling of the second kind for the carbon, oxygen, and calcium targets. We restrict our discussion to the EDSV (left panel) and RPWIA (right panel) approaches. Each panel contains the separate transverse $T$ (upper curves) and interference axial-vector transverse $T'$ (lower curves) scaling functions. All results correspond to $q = 1$ GeV/c. Notice that scaling of the second kind, i.e., independence of nuclear targets, works rather well in all cases, namely, the results for different nuclei are located in a very narrow scaling function. In RPWIA scaling of the second kind works extremely well for $f_{T'}$, with the maximum being slightly below 0.8. Notice that the results corresponding to the three nuclear systems collapse into a unique curve. For the other cases, scaling is broken at some level, mainly due to the contribution ascribed to calcium. In summary, scaling of the second kind works better for RPWIA. By contrast, EDSV shows more uncertainty, with $f_T$ and $f_{T'}$ having their maxima significantly higher than the RFG value, 0.75. Notice that $f_T$ reaches its maximum value close to 1. This makes an important difference with the electromagnetic transverse response [39], and it is related to the role that potentials play in the separate vector and axial contributions in the weak responses.

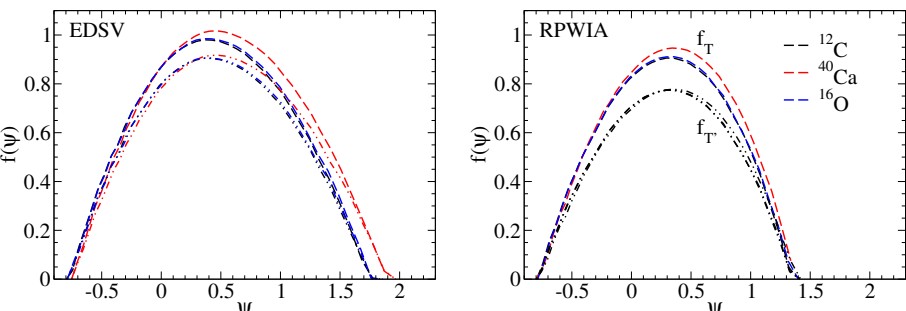

**Figure 6.** Analysis of 2nd kind scaling in carbon, oxygen, and calcium for the transverse (dashed lines) and interference vector-axial transverse (double-dot-dashed lines) scaling functions at $q = 1$ GeV/c for the EDSV (**left** panel) and RPWIA (**right** panel) models.

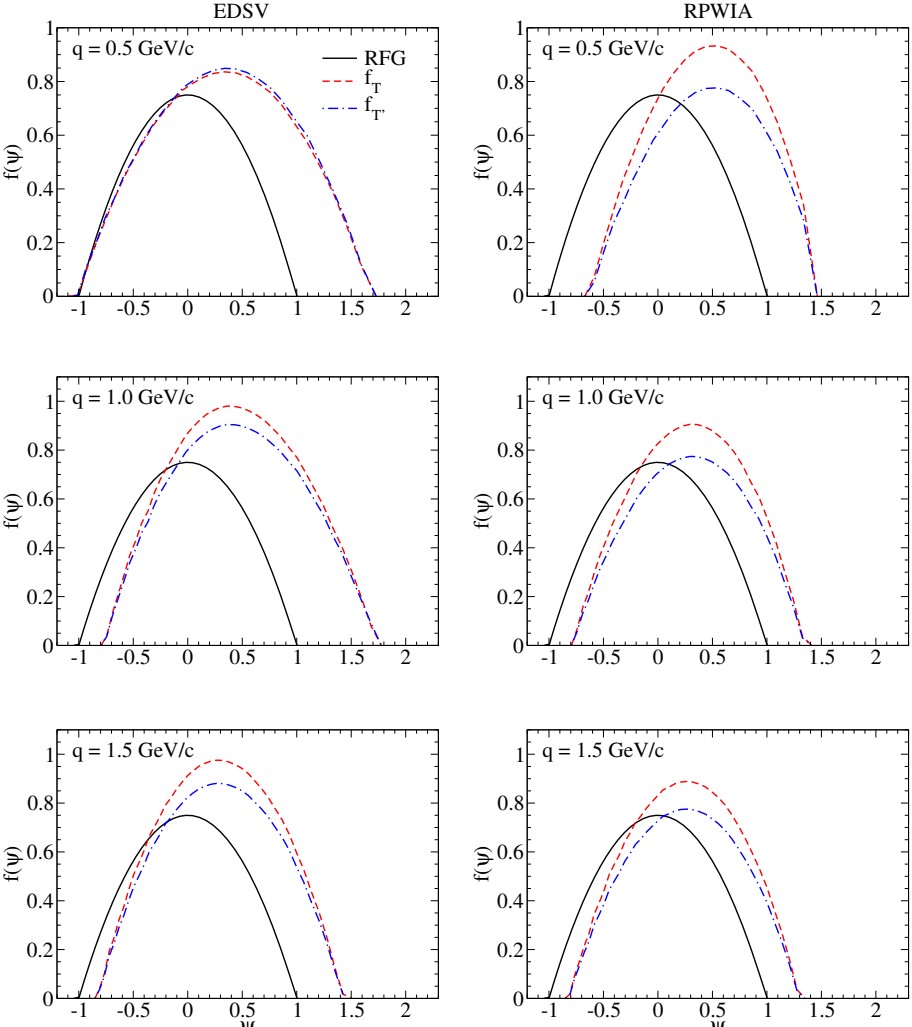

**Figure 7.** Comparison of the $^{12}$C scaling functions $f_T$ and $f_{T'}$ for the EDSV (**left** panels) and RPWIA (**right** panels) models and three different values of the momentum transfer. The RFG scaling function is shown as reference.

For completeness, we compare in Figure 7 $f_T$ and $f_{T'}$ for the two models considered, EDSV and RPWIA, and three different values of the momentum transfer. The RFG prediction is also shown as a reference. Apart from the general shift to larger $\psi$-values shown

by EDSV and RPWIA, in what follows we restrict our attention to the basic differences between the predictions of the two approaches. As observed, the relative discrepancy between $f_T$ and $f_{T'}$ is larger for RPWIA. In contrast, both scaling functions coincide for the EDSV model at $q = 0.5\,\mathrm{GeV/c}$. Regarding the RFG, the EDSV and RPWIA scaling functions achieve their maxima at significantly higher values. Only for RPWIA at $q = 1.5\,\mathrm{GeV/c}$ and $f_{T'}$ (right-bottom panel) is the maximum in the scaling function consistent with RFG. In all remaining cases, and particularly for the EDSV model, the $f_T$ and $f_{T'}$ functions most likely exceed the RFG result. This makes an important difference with the electromagnetic $f_T$ function that is rather similar to the RFG one for the two EDSV and RPWIA models. In both cases, the electromagnetic scaling functions reach their maxima at $f_T(\psi) \approx 0.75$, while the weak transverse responses get close to 1 in some cases, that is, $\sim 25\%$ higher. This particular result, which is not observed with the RMF model applied to finite nuclei [26,30], is probably connected with the assumptions of the RMF model applied to nuclear matter in addition to the specific role played by the axial term in the weak transverse responses.

## 4. Conclusions

This paper closely follows the study already performed for the electromagnetic quasielastic $(e, e')$ response functions presented in [39]. Here, we extend our investigation to the case of charged-current quasielastic neutrino-nucleus scattering within the relativistic mean field applied to nuclear matter. We analyze the five weak response functions that enter in $(\nu_\ell, \ell)$ reactions exploring the effects of the scalar and vector potentials introduced in the initial and/or final nucleon states. Different options have been considered: from the most general case with energy/momentum dependent potentials in both initial and final states to more simple situations with fixed constant potentials equal in both nucleon states, the plane wave limit, that is, no potentials in the final states, and the pure relativistic Fermi gas model taken as reference.

The relativistic mean field model applied to finite nuclei has proved its capability to successfully describe the electromagnetic and weak responses for a large variety of kinematical situations. However, it fails at very high values of the momentum transfer because of the strong energy-independent potentials involved. Here, following the work carried out in [39], we use a simpler, oversimplified description of the scattering processes, but it provides analytical expressions from which the effects ascribed to the main ingredients in the problem are significantly clarified, for instance, how the specific shape of the potentials affects the weak responses. As expected, the results corresponding to energy-dependent potentials in the initial and final states approach the ones in the plane wave limit. However, very significant differences emerge in comparison with the electromagnetic responses. In contrast to $(e, e')$, where the discrepancy in the maximum between the EDSV/RPWIA and RFG predictions is not too large, for neutrinos the differences are much bigger. This is more clearly illustrated when comparing the results for the various scaling functions and is related to the role of the potentials in the axial sector.

From the study of scaling and supercaling one concludes that scaling of the first kind works reasonably well in some kinematical situations, although the strength of the functions (for the different models) departs significantly from the RFG prediction. This makes a crucial difference with the electromagnetic responses. With regard to scaling of the second kind, it works extremely well for the two weak transverse responses, but again with values at the maxima much higher than the RFG one in most of the cases.

In summary, in this work we have extended to the weak sector our previous investigations on pure electromagnetic responses. Using a simple but fully relativistic approach, we have explored in detail the behavior of the responses and scaling/superscaling functions with emphasis in the role played by the relativistic scalar and vector potentials and their effects in the observables. Particular interest has been paid to the visible differences emerged in both the pure electromagnetic and weak nuclear responses.

**Author Contributions:** Conceptualization, S.C.-B., J.A.C. and G.D.M.; methodology, S.C.-B. and J.A.C.; software, S.C.-B.; validation, S.C.-B., J.A.C. and G.D.M.; formal analysis, S.C.-B.; writing—original draft preparation, S.C.-B., J.A.C. and G.D.M.; writing—review and editing, S.C.-B., J.A.C. and G.D.M.; visualization, S.C.-B. and G.D.M. All authors have read and agreed to the published version of the manuscript.

**Funding:** This work has been partially supported by the Spanish Ministerio de Ciencia e Innovación and ERDF (European Regional Development Fund) under Contract No. PID2020-114687GB-100, by the Junta de Andalucia (FQM 160, SOMM17/6105/UGR and P20-01247); it is supported in part by the University of Tokyo ICRR's Inter-University Research Program FY2022 (Ref. 2022i-J-001) & FY2023 (Ref. 2023i-J-001).

**Data Availability Statement:** Not applicable.

**Conflicts of Interest:** The authors declare no conflict of interest. They have no known competing financial interests or personal relationships that could have appeared to influence the work reported in this paper.

## Abbreviations

The following abbreviations are used in this manuscript:

| | |
|---|---|
| CP | Charge-Parity |
| QE | Quasielastic |
| RFG | Relativistic Fermi Gas |
| RMF | Relativistic Mean Field |
| RPWIA | Relativistic Plane Wave Impulse Approximation |
| DIS | Deep Inelastic Scattering |
| MEC | Meson Exchange Currents |
| EDSV | Energy-Dependent Scalar Vector |
| CtSV | Constant Scalar Vector |
| PCAC | Partially Conserved Axial Current |
| CC | Charge–Charge |
| CL | Charge–Longitudinal |
| LL | Longitudinal–Longitudinal |
| T | Transverse |
| T′ | Transverse interference |
| SuSAv2 | SuperScaling Approach version 2 |
| MiniBooNE | Mini Booster Neutrino Experiment |
| MINERvA | Main Injector Neutrino ExpeRiment to study v-A interactions |
| MicroBooNE | Micro Booster Neutrino Experiment |
| T2K | Tokai to Kamioka experiment |
| DUNE | Deep Underground Neutrino Experiment |

## Appendix A. Single-Nucleon Tensor $T^{\mu\nu}$

The general expression for the weak single-nucleon tensor in the most general case, i.e., different scalar and vector relativistic potentials in the initial and final states, can be written in the form:

$$
T^{\mu\nu} = -\frac{(M_f^* + M_i^*)^2}{2}\left(g^{\mu\nu} - \frac{Q^{*\mu}Q^{*\nu}}{Q^{*2}}\right)\left[\tilde{\tau}^*\tilde{G}_M^{*2} + G_A^2\left(1 + \tilde{\tau}^*\right)\right]
$$

$$
+ \left(2P_i^{*\mu}P_i^{*\nu} + P_i^{*\mu}Q^{*\nu} + Q_i^{*\mu}P_i^{*\nu} + \frac{Q^{*\mu}Q^{*\nu}}{2}\right)\left[\frac{\tilde{\tau}^*\tilde{G}_M^{*2} + \tilde{G}_E^{*2}}{\left(1 + \tilde{\tau}^*\right)} + G_A^2\right]
$$

$$
+ \frac{Q^{*\mu}Q^{*\nu}}{2\tilde{\tau}^*}\left(G_A - \tilde{\tau}^*\tilde{G}_P^*\right)^2 + 2i\epsilon^{\mu\nu\beta\lambda}P_{i_\beta}^*\left(Q_\lambda^* + \Delta V_\lambda\right)G_A\tilde{G}_M^*
$$

$$
- \frac{g^{\mu\nu}}{2}(M_f^* - M_i^*)^2\tilde{G}_M^{*2} + \frac{Q^{*\mu}Q^{*\nu}}{2}\left(\frac{M_f^* - M_i^*}{M_f^* + M_i^*}\right)^2\tilde{G}_P^{*2}
$$

$$
+ \left(P_i^{*\mu}Q^{*\nu} + Q^{*\mu}P_i^{*\nu} + Q^{*\mu}Q^{*\nu}\right)\frac{(M_f^* - M_i^*)}{M_f^* + M_i^*}\left[\frac{\tilde{G}_M^*(\tilde{G}_M^* - \tilde{G}_E^*)}{1 + \tilde{\tau}^*} + G_A\tilde{G}_P^*\right]
$$

$$
+ \Delta V\left\{g^{\mu\nu}\left[\left(2\tilde{\lambda}^*M_i^* - \tilde{\epsilon}_i^*(M_f^* - M_i^*)\right)\frac{\tilde{G}_M^*(\tilde{G}_M^* - \tilde{G}_E^*)}{1 + \tilde{\tau}^*}\right.\right.
$$

$$
- \frac{\Delta V}{2}\left(\tilde{\epsilon}_i^{*2} + 2\tilde{\lambda}^*\tilde{\epsilon}_i^* - (1 + \tilde{\tau}^*)\right)\frac{(\tilde{G}_M^* - \tilde{G}_E^*)^2}{\left(1 + \tilde{\tau}^*\right)^2}\Bigg]
$$

$$
+ \left[-\frac{\tilde{\lambda}^*}{M_f^* + M_i^*}\left(4P_i^{*\mu}P_i^{*\nu} + P_i^{*\mu}Q^{*\nu} + Q^{*\mu}P_i^{*\nu}\right)\right.
$$

$$
+ \frac{\tilde{\epsilon}_i^*}{M_f^* + M_i^*}\left(P_i^{*\mu}Q^{*\nu} + Q^{*\mu}P_i^\nu + Q^{*\mu}Q^{*\nu}\right)
$$

$$
- \frac{\Delta V}{(M_f^* + M_i^*)^2}\left(2P_i^{*\mu}P_i^{*\nu} + P_i^{*\mu}Q^{*\nu} + Q^{*\mu}P_i^{*\nu}\right)\Bigg]\frac{(G_M^* - G_E^*)^2}{\left(1 + \tilde{\tau}^*\right)^2}\Bigg\}
$$

$$
+ \frac{(P_i^{*\mu}\Delta V^\nu + \Delta V^\mu P_i^{*\nu})}{M_f^* + M_i^*}\left[(M_f^* - M_i^*)\left(\frac{G_M^*(\tilde{G}_M^* - \tilde{G}_E^*)}{1 + \tilde{\tau}^*} + G_A\tilde{G}_P^*\right)\right.
$$

$$
+ \Delta V\left(\tilde{\epsilon}_i^* + \tilde{\lambda}^*\right)\frac{(\tilde{G}_M^* - \tilde{G}_E^*)^2}{(1 + \tilde{\tau}^*)^2}\Bigg]
$$

$$
+ \left(Q^{*\mu}\Delta V^\nu + \Delta V^\mu Q^{*\nu}\right)\left[\frac{-M_i^*}{(M_f^* + M_i^*)}\left[\frac{\tilde{G}_M^*\left(\tilde{G}_M^* - \tilde{G}_E^*\right)}{1 + \tilde{\tau}^*} + G_A\tilde{G}_p^*\right]\right.
$$

$$
+ \frac{\Delta V}{2(M_f^* + M_i^*)}\tilde{\epsilon}_i^*\frac{(\tilde{G}_M^* - \tilde{G}_E^*)^2}{(1 + \tilde{\tau}^*)^2} + \frac{\tilde{G}_P^{*2}}{2}\left(\frac{(M_f^* - M_i^*)^2}{(M_f^* + M_i^*)^2} + \tilde{\tau}^*\right)\Bigg]
$$

$$
+ \frac{\Delta V^\mu \Delta V^\nu}{2}\left[-\left(1 + \tilde{\tau}^*\right)\frac{(\tilde{G}_M^* - \tilde{G}_E^*)^2}{(1 + \tilde{\tau}^*)^2} + \tilde{G}_P^{*2}\left(\frac{(M_f^* - M_i^*)^2}{(M_f^* + M_i^*)^2} + \tilde{\tau}^*\right)\right].
$$

$$
(A1)
$$

The single nucleon form factors in (A1) are defined as:

$$\tilde{G}_M^* = F_1 + \tilde{F}_2^*; \quad \tilde{G}_E^* = F_1 - \tau^* \tilde{F}_2^*; \quad \tilde{G}_P^* = \frac{\left(M_f^* + M_i^*\right)}{2M} G_P \tag{A2}$$

with $\tilde{F}_2^* \equiv \frac{\left(M_f^* + M_i^*\right)}{2M} F_2$, and the following set of dimensionless variables have been introduced [24,25,47]:

$$
\begin{aligned}
&\kappa \equiv \frac{q}{2M}; \qquad \lambda \equiv \frac{\omega}{2M}; \qquad \eta \equiv \frac{p}{M}; \qquad \tau = \kappa^2 - \lambda^2 \\
&s_m \equiv \frac{S(\mathbf{p})}{M}; \qquad \Delta v \equiv \frac{\Delta V}{2M}; \qquad \lambda^* \equiv \frac{\omega^*}{2M} = \lambda - \Delta v; \qquad \tau^* = \kappa^2 - \lambda^{*2} \\
&\epsilon^* \equiv \frac{E_{\mathbf{p}}^*}{M} = \sqrt{\eta^2 + (1+s_m)^2}; \quad \Delta m^{*2} \equiv \frac{1}{4}\left[\left(\frac{M_f^*}{M}\right)^2 - \left(\frac{M_i^*}{M}\right)^2\right]; \quad \rho^* = \left(1 + \frac{\Delta m^{*2}}{\tau^*}\right) \\
&\tilde{\kappa}^* \equiv \frac{q}{(M_f^* + M_i^*)}; \qquad \tilde{\lambda}^* \equiv \frac{\omega^*}{(M_f^* + M_i^*)}; \qquad \tilde{\tau}^* = \tilde{\kappa}^{*2} - \tilde{\lambda}^{*2} \\
&\tilde{\eta}_{i,f}^* \equiv \frac{p_{i,f}}{(M_f^* + M_i^*)/2}; \qquad \tilde{\epsilon}_{i,f}^* \equiv \frac{E_{i,f}^*}{(M_f^* + M_i^*)/2}.
\end{aligned}
\tag{A3}
$$

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
