# Peer review of "Weak Neutrino (Antineutrino) Charged-Current Responses and Scaling for Nuclear Matter in the Relativistic Mean Field"

_universe, doi:10.3390/universe9050240_

Round 1

Reviewer 1 Report

This is a fine study of a very important subject (modeling of CC neutrino reactions at high energies).  I recommend that the paper be accepted as written.

Author Response

We appreciate very much the general comment of the Referee.

Reviewer 2 Report

This manuscript investigates the charged-current quasi-elastic neutrino-nucleus scattering within the relativistic mean field theory. The weak responses and scaling functions are investigated in detail with different
approximations. The manuscript contains a detailed examination and important results.
I recommend the publication of this manuscript in the Journal of Universe
after taking into account the following minor issues:

* Introduction is well written. I suggest adding the motivation to use the nuclear target to complete the introduction.

* This work studies the nuclear target of carbon (12C), oxygen (16O), and calcium (40Ca) with the nuclear matter application. It is very questionable to assume that these small targets are approximated by infinite matter.
How do you determine the values of the Fermi momentum for 12C, 16O, and 40Ca? Is the Only difference among different nuclei the Fermi momentum and 'N'? If so, what justifies this simplification?

* SuSAv2  MiniBooNE, MINERvA, MicroBooNE, T2K, DUNE should be in the list of abbreviations.

* line 55:  the scalar and vector potentials -> the scalar (S) and vector (V) potentials

* line 87 What do you mean by 'protons(neutrons)'? Does 'N' mean the number of nucleons? So you do not need to distinguish between protons and neutrons?

* In Fig.7, which target nucleus is used?

Author Response

We attach file with our responses to Referee 2

Reviewer 3 Report

Dear Authors,

This paper reports on explaining the weak neutrino charged current responses and scaling for nuclear matter in the RMF, which is a useful addition to the literature. The paper is generally well written, however here are some minor comments :

Please clarify the following arguments with perhaps a suitable rephrasing of the following  line numbers.

1. Introduction:

  1.  Line 40 : What is SuSAv2 approach? Can you please give any reference?

2. General formalism:

  1. Line 79-85: You mentioned C, L and T, but what is Tprime? What is R? Can you please explain it a little more clearly?

  2. Line 125-129: You put the values for the parameters a_i and b_i as well as K_F^0, Can you please address any references from where you get those values?

3. Discussion of results:

  1. Fig 1 and Fig 2: Line 187-200:  It is a little unclear the “reason” why you are seeing this behavior (significant differences between various models as you change the value of q)?

  2. Fig 3 and 4: Why don’t you include R_CL or R_LL here? Any reason?

Thank You

Author Response

We include file with our responses to Referee 3
